**Subject Category:**
Biology (whole organism)

biomathematics/ecology/mathematical modelling

population dynamics, Hassell model,
first-principles derivation, individual,
contest competition, scramble competition

**Author for correspondence:**
Masahiro Anazawa
e-mail: anazawa@tohtech.ac.jp

# Inequality in resource allocation and population dynamics models

## Masahiro Anazawa

Department of Environment and Energy, Tohoku Institute of Technology, Sendai 982-8577, Japan

MA, 0000-0001-7257-6103

The Hassell model has been widely used as a general discrete-time population dynamics model that describes both contest and scramble intraspecific competition through a tunable exponent. Since the two types of competition generally lead to different degrees of inequality in the resource distribution among individuals, the exponent is expected to be related to this inequality. However, among various first-principles derivations of this model, none is consistent with this expectation. This paper explores whether a Hassell model with an exponent related to inequality in resource allocation can be derived from first principles. Indeed, such a Hassell model can be derived by assuming random competition for resources among the individuals wherein each individual can obtain only a fixed amount of resources at a time. Changing the size of the resource unit alters the degree of inequality, and the exponent changes accordingly. As expected, the Beverton–Holt and Ricker models can be regarded as the highest and lowest inequality cases of the derived Hassell model, respectively. Two additional Hassell models are derived under some modified assumptions.

## 1. Introduction

The population dynamics of seasonally reproducing species with non-overlapping generations, such as insects, are often described using discrete-time population models $N_{t+1} = f(N_t)$, which express the population size at one generation $N_{t+1}$ as a function of the population size at the previous generation $N_t$. Classic examples include the Beverton–Holt model [1], Ricker model [2], Hassell model [3] and Maynard-Smith–Slatkin model [4]. However, these models were originally introduced as phenomenological models at the population level and were not derived from more fundamental processes such as the interactions among individuals. Therefore, the individual-level mechanisms underlying these models are not very clear. However, in recent years, research to derive these population models from more fundamental processes has progressed [5–8]. If a

**Figure 1.** Reproduction curves of the Hassell model $N_{t+1} = \lambda N_t/(1 + aN_t)^b$ for different values of exponent $b$. $\lambda = 1.5$; $a$ is determined by the condition $N_* = 10$.

population model can be derived from basic processes, such as resource competition among individuals, we can better understand the individual-level processes and phenomena underlying the model and clarify how the parameters of the population model are related to the parameters characterizing the corresponding individual-level processes.

The Hassell model

$$N_{t+1} = \frac{\lambda N_t}{(1 + aN_t)^b} \tag{1.1}$$

can describe various types of density dependence by changing the exponent $b$, and thus it has been widely used as a general population model [3,9]. When $b = 1$, the model gives a curve that monotonically increases towards a maximum, showing exactly compensating density dependence for large population sizes (figure 1). When $b > 1$, the model gives overcompensating curves, which increase to a maximum and then decrease with increasing population size. The degree of overcompensation grows as $b$ increases. The density-dependence types of exact compensation and overcompensation are often supposed to reflect different types of intraspecific competition. Nicholson [10] emphasized two extreme types of intraspecific competition, which he called contest and scramble competition. In ideal contest for resources, a few successful individuals obtain their full requirements for survival and reproduction, but others obtain nothing [3,11]. Consequently, when there are many individuals, the number of successful individuals and their offspring are almost independent of population size, and the population dynamics describe an exact compensating curve. By contrast, in ideal scramble competition, all individuals are assumed to scramble for the resources and receive an equal share of the resources [3,11]. In a large population, most individuals cannot obtain sufficient resources for their survival and reproduction, so the dynamics describe an overcompensating curve.

Hassell [3] remarked that in his model, $b = 1$ corresponds to ideal contest competition, the limit $b \to \infty$ corresponds to ideal scramble competition and $b > 1$ corresponds to varying combinations of scramble and contest. The monopolization of resources by a few individuals in ideal contest can be considered as an unequal distribution of resources [12,13]. Accordingly, exponent $b$ is expected to be related to the degree of inequality in resource distribution among the individuals. As $b$ enlarges, the degree of inequality should decrease. Although the Hassell model was originally introduced as a phenomenological model at the population level, several authors have succeeded in deriving it from first principles. However, as described below, none of these derivations are consistent with the expected relationship between exponent $b$ and the inequality in resource allocation.

Existing first-principles derivations of the Hassell model can be broadly classified into two approaches. The first approach considers population dynamics in an environment comprising many resource patches. De Jong [14,15] showed that when the individuals are distributed negative binomially over the patches, discrete-time population dynamics are described by a Hassell model. However, the exponent of her model is $b = k + 1$, where $k$ is a clumping parameter of the negative binomial distribution, and hence it is not related to the inequality in resource allocation. In similar ways, various discrete-time population models [6,7,16] and interspecific competition models [17,18] have been derived from first principles. The second approach assumes a system of differential equations describing continuous-time dynamics within a year, from which a discrete-time model for the between-year dynamics is derived. It

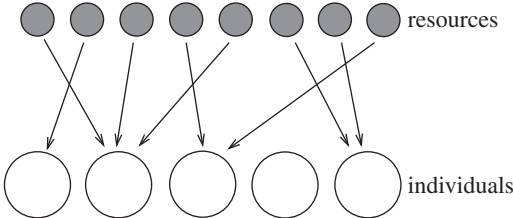

**Figure 2.** Competition for limited resources among several individuals. If each individual can obtain only a fixed amount of resources at a time, it follows that the resources are essentially the same as being divided into many pieces of this fixed size, which are distributed among the individuals.

is well known that when a population is subject to mortality that linearly increases with density, the population size after a certain period is described by a Beverton–Holt model as a function of the original population size. Extending this idea, Nedorezov *et al.* [19] derived a Hassell model by assuming discrete reproduction at the end of the year with fecundity exponentially decreasing with the mean population size over the year. Further, Hassell models have also been derived from various continuous-time systems incorporating interactions between species or age classes [5,20–23]. However, similar to de Jong's model, the exponents of these Hassell models are not related to the inequality among individuals. In summary, although the Hassell model has been derived in various ways, no existing derivation has captured the expected relationship between the exponent and the inequality in resource allocation.

This paper explores whether a Hassell model whose exponent is related to the inequality can be derived from first principles. Such a Hassell model can indeed be derived by assuming random competition for resources among the individuals wherein each individual can obtain only a fixed amount of resources at a time. The Beverton–Holt and Ricker models can be regarded as special cases of the derived Hassell model, namely, the models for the highest and lowest inequality, respectively. Furthermore, two additional Hassell models are derived when some assumptions are modified.

# 2. Derivation of population models

## 2.1. Hassell model

This section aims to derive a Hassell model whose exponent is related to the degree of inequality in resource distribution among the individuals of a population. The inequality can arise from various causes, e.g. different competitiveness levels among individuals, but here it is attributed solely to the randomness in resource distribution among the individuals. We show that the Hassell model describes the expected value of the population size $N_{t+1}$ at generation $t+1$ as a function of $N_t$.

Consider a situation in which $N$ individuals compete for limited resources. Let us assume that each individual can obtain only a certain fixed amount $u$ of resources at a time. It follows that the resources, which may be continuous or discrete, are effectively the same as being divided into many pieces of this fixed size (resource units), which are collected by the competing individuals (figure 2). Assuming that there are $M$ resource units in total (i.e. the total resource amount is $R = uM$) and that each resource unit is consumed by a randomly selected individual, the total number of resource units obtained by a given individual after the competition follows a binomial distribution. More specifically, the probability $p_m$ of a given individual obtaining exactly $m$ resource units is

$$p_m(M) = \left(\frac{1}{N}\right)^m \left(1 - \frac{1}{N}\right)^{M-m} \binom{M}{m}, \tag{2.1}$$

where $m \leq M$. The variance of the actual amount of resources allocated to an individual $um$ is then given by

$$\mathrm{Var}[um] = uR\left(1 - \frac{1}{N}\right)\frac{1}{N}. \tag{2.2}$$

Note that when $R$ is fixed, this variance is proportional to the unit size $u$. This shows that the inequality among individuals increases with unit size. Assuming that an individual requires at least $s$ resource units to reproduce, the probability that a given individual successfully reproduces is

$$Q(M) = \sum_{m=s}^{M} p_m(M). \tag{2.3}$$

Letting $f(N)$ denote the expected population at the next generation, the population model is then written as

$$f(N) = \lambda N Q(M),\tag{2.4}$$

where $\lambda$ is the expected number of offspring produced by a reproductive individual.

When the value of $R$ is known, equation (2.4) gives the expected population at the next generation. However, unfortunately, this equation does not give a Hassell model. Then, let us consider a situation where the value of $R$ is unknown and only its probability distribution is known. In this case, the expected population that we can infer based on the information we have is not described by equation (2.4) but determined by combining it with the probability distribution of $R$. As a specific distribution of $R$, we choose an exponential distribution with the probability density

$$q(R) = \frac{e^{-R/\bar{R}}}{\bar{R}},\tag{2.5}$$

where $\bar{R}$ is the expected value of $R$ (the physical meaning of this distribution will be discussed later). The total number $M$ of resource units, i.e. the integer part of $R/u$, follows a geometric distribution

$$P_M = (1 - e^{-u/\bar{R}})\,e^{-Mu/\bar{R}}.\tag{2.6}$$

In this situation, the population model is given by averaging the right-hand side of equation (2.4) with this distribution as

$$f(N) = \sum_{M=0}^{\infty} \lambda N Q(M) \cdot P_M.\tag{2.7}$$

Interestingly, this infinite sum proves to be the following Hassell model (see appendix A.1 for details):

$$f(N) = \frac{\lambda N}{[1 + (e^{u/\bar{R}} - 1)N]^s}.\tag{2.8}$$

The exponent $s$ in this model can be written as $s'/u$, where $s'$ is the actual resource amount required for reproduction. This shows that the exponent is inversely proportional to $u$. As mentioned before, $u$ is related to the inequality in resource allocation, so the exponent is indeed related to the inequality. A larger exponent implies less inequality in resource allocation. Therefore, we have succeeded in deriving a Hassell model consistent with the expected relationship between the exponent and the inequality in resource allocation. Note that the model (2.8) predicts the expected, rather than the actual, population size in the next generation, so that it cannot be used iteratively to predict population sizes in the subsequent generations.

## 2.2. Beverton–Holt and Ricker models

Let us consider the largest and smallest inequalities in the derived Hassell model (2.8). When the unit size $u$ is considerably large, an individual can reproduce after consuming only one resource unit, which means $s = 1$. In this case, the allocation is maximally unequal, and the model (2.8) becomes the following Beverton–Holt:

$$f(N) = \frac{\lambda N}{1 + (e^{u/\bar{R}} - 1)N}.\tag{2.9}$$

By contrast, the allocation is equalized in the limit $u \to 0$ with $s' = us$ fixed ($s = s'/u \to \infty$). In this limit, the resources are equally partitioned among all individuals, and equation (2.8) becomes the Ricker model

$$f(N) = \lambda N \exp\left(-\frac{s'}{\bar{R}}N\right).\tag{2.10}$$

Therefore, the Beverton–Holt and Ricker models can both be interpreted as special cases of the Hassell model (2.8) in which the allocation is maximally unequal and completely equal, respectively.

Figure 3 shows curves of the Hassell model (2.8) for several values of $s$. As $s$ increases ($u$ decreases), the degree of inequality decreases, increasing the degree of overcompensation.

## 2.3. Interpretation of the derivation

When deriving the Hassell model (2.8), we assumed that the total resource amount $R$ follows the exponential distribution (2.5). This assumption needs to be interpreted. Consider a continuous random variable with an unknown probability distribution. The distribution that maximizes information

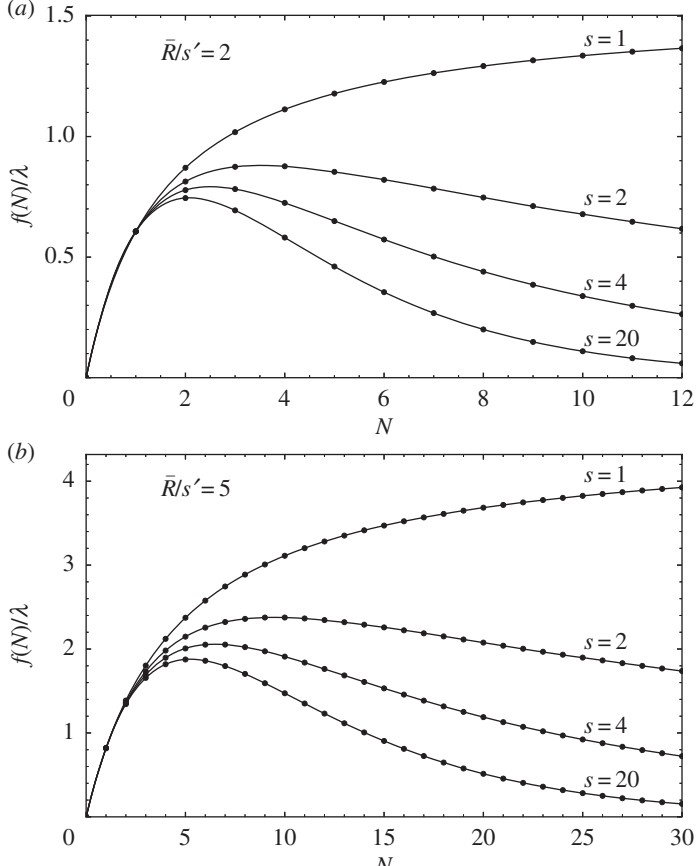

**Figure 3.** Reproduction curves of the derived Hassell model (2.8) for different values of $s$: (a) $\bar{R}/s' = 2$ and (b) $\bar{R}/s' = 5$. As $s$ increases (the resource unit size $u$ decreases), the degree of inequality in the allocated resources decreases, and the curves develop higher degrees of overcompensation.

entropy is the least-biased prediction of the distribution consistent with the prior knowledge, which describes the most natural prediction [24,25]. The distribution that maximizes information entropy depends on the prior knowledge we have. For example, when we know the range of realized values, the distribution that is both consistent with this prior knowledge and maximizes information entropy is the uniform distribution whose domain is identical to that range. When only the expected value of the variable instead of the range is known, the probability distribution that maximizes information entropy turns out to be an exponential distribution [24,25]. In this interpretation, the Hassell model (2.8) represents the most natural estimate of the expected number of individuals in the next generation when only the expected amount of total resources is known.

Another possible interpretation of the exponential distribution of $R$ is that it represents random temporal variation of $R$. When we do not know the value of $R$ itself but know its probability distribution, the Hassell model (2.8) represents the expected value of population size at the next generation that can be inferred based on the knowledge we have. Alternatively, when we know the varying value of $R$ at each generation, if $R$ follows the exponential distribution, the Hassell model (2.8) is interpreted as the mean relationship averaged over time between $N_t$ and the expected value of $N_{t+1}$.

# 3. Extended models

## 3.1. Non-constant resource threshold

In §2, we assumed that all individuals need the same number $s$ of resource units for reproduction. However, in many real populations, $s$ should vary among the individuals. The following discussion considers population models with variable reproduction requirements. Interestingly, under certain conditions, we can obtain Hassell models that include the effect of $s$ variation among individuals in their exponents.

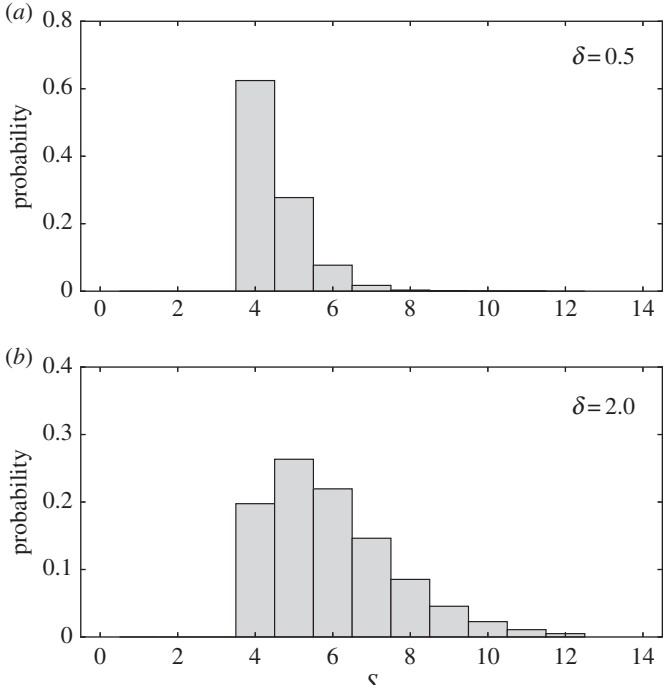

**Figure 4.** Examples of the distribution of $s$ defined by equation (3.2) for (a) $\delta = 0.5$ and (b) $\delta = 2.0$. The variance enlarges as $\delta$ increases. $k = s_0 = 4.0$.

When $s$ varies among the individuals, the expected number of individuals in the next generation can be written as

$$f(N) = \sum_{s=0}^{\infty} \lambda N [1 + (e^{u/\bar{R}} - 1)N]^{-s} p_s, \tag{3.1}$$

where $p_s$ is the probability of a given individual requiring $s$ resource units for reproduction. As an example, let us consider a case in which $p_s = 0$ for $s$ less than a positive integer $s_0$, and for $s \geq s_0$, $s - s_0$ follows a negative binomial distribution

$$p_s = \frac{\Gamma(s - s_0 + k)}{\Gamma(k)\Gamma(s - s_0 + 1)} \left(\frac{\delta}{k}\right)^{s-s_0} \left(1 + \frac{\delta}{k}\right)^{-s+s_0-k}, \tag{3.2}$$

where $\delta = \bar{s} - s_0 \, (> 0)$, $\bar{s}$ is the mean of $s$ and $k \, (> 0)$ is a shape parameter (figure 4). The variance of $s$ is $\sigma_s^2 = \delta + \delta^2/k$, so it grows with increasing $\delta$ or decreasing $k$. In the limit $\delta \to 0$, the variance vanishes, recovering the constant $s$ in §2. Substituting equation (3.2) into equation (3.1) and summing the terms gives the following population model (see appendix A.2 for detail):

$$f(N) = \frac{\lambda N}{[1 + (e^{u/\bar{R}} - 1)N]^{s_0}} \left(\frac{1 + (e^{u/\bar{R}} - 1)N}{1 + (1 + \delta/k)(e^{u/\bar{R}} - 1)N}\right)^k. \tag{3.3}$$

Interestingly, after imposing an additional condition $k = s_0$, this model transforms into a Hassell model

$$f(N) = \frac{\lambda N}{[1 + (1 + \delta/k)(e^{u/\bar{R}} - 1)N]^k}. \tag{3.4}$$

Even under this condition, the variance of $s$ can be changed freely by varying $\delta$ (figure 4). As evidenced below, the exponent $k$ of this model includes the effect of varying $s$ among the individuals. From $\sigma_s^2 = \delta(1 + \delta/k)$ and $\bar{s} = k + \delta$, we have

$$k = \frac{\bar{s}}{1 + \sigma_s^2/\bar{s}} = \frac{\bar{s}'}{u + \sigma_{s'}^2/\bar{s}'} \tag{3.5}$$

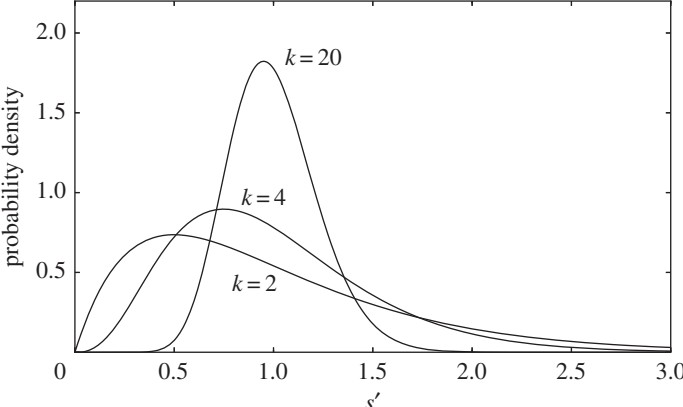

**Figure 5.** Examples of the gamma distribution (3.8). The negative binomial distribution (3.2) transforms to this distribution in the limit $u \to 0$. The variance decreases with increasing $k$. $\bar{s}' = 1$.

and

$$\delta = \frac{\sigma_s^2}{1 + \sigma_s^2/\bar{s}} = \frac{\sigma_{s'}^2/u}{u + \sigma_{s'}^2/\bar{s}'}, \tag{3.6}$$

where $\bar{s}'$ and $\sigma_{s'}^2$ are, respectively, the mean and variance of $s' = us$, i.e. the actual resource amount required for reproduction. Note that the exponent $k$ in equation (3.5) depends on $\sigma_{s'}^2$ and the unit size $u$. Therefore, it includes the effects of varying $s'$ along with the inequality in the allocated resources. Increasing $\sigma_{s'}^2$ decreases the exponent $k$, thereby reducing the level of overcompensation of the reproduction curve. This is probably because a large variation in $s'$ suppresses an abrupt increase in individuals that fail to reproduce when the population size increases.

As shown in equation (3.5), due to the $\sigma_{s'}^2$ term in the denominator, the exponent of model (3.4) does not approach infinity in the limit $u \to 0$, unlike Hassell model (2.8). Accordingly, even in the limit of equal distribution of resources, the model remains a Hassell model as follows:

$$f(N) = \frac{\lambda N}{[1 + \bar{s}'/(k\bar{R}) \cdot N]^k}. \tag{3.7}$$

In this case, the exponent $k = \bar{s}'^2/\sigma_{s'}^2$ includes only the effect of the variation in $s'$. Note that the same model can be obtained from model (3.3), i.e. the model before imposing the condition $k = s_0$, by taking the limit $u \to 0$ with $\bar{s}' = u\bar{s}$ fixed. Accordingly, the exponent must be a positive integer in model (3.4) but can be any positive real number in model (3.7). It is also worth noting that in this limit, the distribution (3.2) transforms into a continuous one, specifically a gamma distribution with mean $\bar{s}'$, shape parameter $k$ ($>0$) and probability density

$$q(s') = \frac{k^k}{\Gamma(k)\bar{s}'} \left(\frac{s'}{\bar{s}'}\right)^{k-1} e^{-ks'/\bar{s}'}. \tag{3.8}$$

Here, the variance of $s'$ is $\sigma_{s'}^2 = \bar{s}'^2/k$, which decreases as $k$ increases, sharpening the peak in the distribution (figure 5). Thus, equation (3.7) can be considered as the population model when $s'$ follows the above gamma distribution and the resources are distributed equally.

## 3.2. Non-constant fecundity

In §2, the fecundity (expected number of offspring produced by an individual) was assumed constant when the obtained resource amount exceeded a threshold. However, this assumption may be overly idealized. In the following discussion, we relax this assumption and consider population models in which the fecundity depends on the amount of allocated resources.

The fecundity should gradually increase towards a maximum as the amount of allocated resources increases. Therefore, we describe the fecundity by the following function of the number $m$ of the

allocated resource units:

$$\lambda_m = \begin{cases} \lambda(1 - e^{-\gamma u(m-s+1)}) & (m \geq s) \\ 0 & (m < s), \end{cases} \tag{3.9}$$

where $s$ is a positive integer and $\lambda$ and $\gamma$ are positive parameters. When $m$ is greater than or equal to $s$, $\lambda_m$ increases towards the maximum $\lambda$ with increasing $m$. In the limit $\gamma \to \infty$, equation (3.9) reverts to the case of §2, with $\lambda_m = \lambda$ for $m \geq s$ and $\lambda_m = 0$ for $m < s$. The expected number of individuals in the next generation can be written as

$$f(N) = N \sum_{m=s}^{\infty} \lambda_m \hat{p}_m, \tag{3.10}$$

where $\hat{p}_m$ is the probability that a given individual obtains exactly $m$ resource units when $R$ follows the exponential distribution (2.5). $\hat{p}_m$ is determined from

$$\hat{p}_m = \sum_{M=0}^{\infty} p_m(M) P_M, \tag{3.11}$$

and a calculation with equations (2.1) and (2.6) yields the following geometric distribution (see appendix A.1):

$$\hat{p}_m = \left( \frac{1}{(e^{u/\bar{R}} - 1)N} \right)^m \left( 1 + \frac{1}{(e^{u/\bar{R}} - 1)N} \right)^{-m-1}. \tag{3.12}$$

Substituting equations (3.12) and (3.9) into equation (3.10), followed by the summation of the terms, the population model can be expressed as

$$f(N) = \frac{\lambda N}{[1 + (e^{u/\bar{R}} - 1)N]^{s-1}[1 + \xi(e^{u/\bar{R}} - 1)N]}, \tag{3.13}$$

where $\xi = (1 - e^{-\gamma u})^{-1}$. On comparing this model with the Hassell model (2.8), we find that $N$ is replaced with $\xi N$ ($> N$) in one of the $s$ multiplicative factors of $[1 + (e^{u/\bar{R}} - 1)N]$ in the denominator of equation (2.8). Accordingly, this model gives values smaller than those given by the Hassell model (2.8), reflecting fecundity (3.9) lower than that assumed in the previous model, particularly when $\gamma$ is small.

In the case of highest inequality ($s = 1$), equation (3.13) becomes the Beverton–Holt model

$$f(N) = \frac{\lambda N}{1 + \xi(e^{u/\bar{R}} - 1)N}. \tag{3.14}$$

Conversely, in the case of equal distribution (in the limit $u \to 0$ with $s' = us$ fixed), equation (3.13) becomes

$$f(N) = \lambda N \exp\left( -\frac{s'}{\bar{R}}N \right) \frac{1}{1 + N/(\gamma\bar{R})}. \tag{3.15}$$

This model has the form of the Ricker model (2.10) multiplied by the growth rate of a Beverton–Holt model. Owing to the multiplicative factor, the values in this model are lower than those in the Ricker model (2.10).

# 4. Discussion

The Hassell model was originally intended to describe both contest and scramble competition by means of a tunable exponent [3]. In this interpretation, the exponent is expected to relate to the degree of inequality in the resource allocation, but no first-principles derivation consistent with this expectation has ever been reported. In this paper, we have shown that such a Hassell model can indeed be derived from first principles. This shows that the original Hassell model is not only a convenient expressive population model but also a model with a clear individual-level background. Changing the size of the resource unit alters the degree of inequality and the exponent changes accordingly. In particular, for the cases of the highest and lowest inequalities, the derived Hassell model becomes a Beverton–Holt model and a Ricker model, respectively. Therefore, the three distinct models of Beverton–Holt, Ricker and Hassell can be understood in a unified way through the size of the resource unit.

As stated in the Introduction section, several first-principles derivations of the Hassell model have been published in the literature. The present derivations do not discredit the earlier derivations but

show that the model can be derived in various situations. This paper emphasizes that only the present derivation of model (2.8) is consistent with the original interpretation of the Hassell model [3]. In Hassell's interpretation, $b = 1$, the limit $b \to \infty$ and $b > 1$ correspond to contest competition, scramble competition and varying combinations of scramble and contest, respectively [3]. In the present derivation, $b = 1$ indeed corresponds to the highest inequality condition of ideal contest. Increasing $b$ reduces the degree of inequality, and the competition approaches the equally distributed resource condition of ideal scramble in the limit $b \to \infty$, again consistent with Hassell's interpretation. The Hassell models derived for patchy habitats describe a very different situation [6,7,14]. In this case, the exponent is related only to the degree of clumping of the individuals over the patches, so increasing the exponent does not alter the degree of inequality or the type of competition at all. In §3, we considered a situation in which the amount of resources $s'$ required for reproduction varies among individuals and derived another Hassell model (3.4). The exponent of this equation is related not only to the inequality in resource allocation but also to the variance of $s'$. This result clearly shows that multiple factors can affect the exponent, providing new insights into the density effects underlying the Hassell model.

This paper assumed a simple resource distribution model and showed that the resource unit size affects the resource inequality and thus the degree of overcompensation of the reproduction curve. Such a relationship between the unit size and the resource inequality is expected not only in this case but also in more general situations. Even in a more complicated model of resource distribution, the degree of resource inequality should depend on the size of the resource unit that an individual can obtain at a time. Therefore, it should be a universal property independent of the details of the distribution of resources that the degree of overcompensation grows with decreasing unit size.

The inequality in this paper was attributed only to the random resource distribution among the individuals. However, other causes of inequality are plausible. For example, suppose that every time an individual obtains a resource unit, its probability of obtaining another resource unit increases. In this circumstance, the inequality would be higher than assumed herein. Another source of inequality is the different inherent competitiveness among the individuals. The present paper focused only on the inequality arising from random resource competition in order to reveal the minimum set of assumptions necessary to derive a Hassell model from the individual level.

The Maynard-Smith–Slatkin model $N_{t+1} = \lambda N_t / (1 + a N_t^b)$ is a classic population model similar to the Hassell model [4]. This model is also expected to describe both contest and scramble competition by changing the exponent $b$, but little is known about its individual-level background. It remains an open question whether this model can also be derived in such a way that the exponent is related to resource inequality.

The exponents of the Hassell models (2.8) and (3.4) are limited to integers, but Hassell models with real number exponents are often used phenomenologically. I attempted to derive real number exponents but succeeded only in model (3.7), which is limited to equal resource distributions. Although this paper assumes that the resource unit size is always constant, relaxing this assumption or taking inequalities caused by causes other than random resource distributions into consideration might allow us to derive Hassell models with real number exponents.

Although this study considered only single-species population dynamics, the extension to two-species cases is straightforward when both species have the same resource unit size, and an interspecific competition model can be derived. In general, however, the unit size should depend on the species. This situation is not easily extendible from the present discussion and requires a more careful analysis. In this case, which is beyond the scope of this research, the competition between species of different competition types could be discussed by changing the unit size and hence the degree of inequality of each species independently.

This paper has shown that the exponent of the Hassell model is indeed related to inequality in resource allocation by deriving Hassell equations from first principles, thereby closing the knowledge gap between the original interpretation of the Hassell model and the previous first-principles derivations of the model. Although these equations were derived from rather idealized assumptions, the basic ideas here will also be useful in considering more complicated situations.

Data accessibility. There are no data associated with this study. It is replicable only by mathematical calculations.

Competing interests. The author declares no competing interests.

Funding. The author received no funding for this study.

Acknowledgements. The author is grateful to Drs Takenori Takada, Kazunori Sato, Matthew Holden and Gabriela Gomes for useful comments and discussions. The author also wishes to thank Dr Juniper L. Simonis and one anonymous reviewer for their valuable reviews of the original manuscript.

# Appendix A

## A.1. Derivation of equation (2.8)

The Hassell model (2.8) can be derived from equation (2.7) as follows. As $p_m(M) = 0$ for $m > M$, equation (2.3) can be rewritten as

$$Q(M) = \sum_{m=s}^{\infty} p_m(M), \tag{A1}$$

and then substituting this equation into equation (2.7) gives

$$f(N) = \lambda N \sum_{M=0}^{\infty} \sum_{m=s}^{\infty} p_m(M) P_M. \tag{A2}$$

Interchanging the order of the two summations, we can rewrite this equation as

$$f(N) = \lambda N \sum_{m=s}^{\infty} \hat{p}_m, \tag{A3}$$

where

$$\hat{p}_m = \sum_{M=0}^{\infty} p_m(M) P_M, \tag{A4}$$

represents the probability of a given individual obtaining exactly $m$ resource units when $M$ follows the geometric distribution (2.6). Substituting equations (2.1) and (2.6) into equation (A 4) and recalling that $p_m(M) = 0$ for $m > M$, we have

$$\hat{p}_m = \sum_{M=m}^{\infty} c\, a^M \left(\frac{1}{N}\right)^m \left(1 - \frac{1}{N}\right)^{M-m} \frac{M!}{m!(M-m)!}, \tag{A5}$$

where $a = e^{-u/\bar{R}}$ and $c = 1 - a$. Relabelling the index $M$ as $M = M' + m$, we have

$$\hat{p}_m = \sum_{M'=0}^{\infty} c\, a^{M'+m} \left(\frac{1}{N}\right)^m \left(1 - \frac{1}{N}\right)^{M'} \frac{(M' + m)!}{m!M'!}. \tag{A6}$$

Applying the expansion formula

$$(1 - x)^{-k-1} = \sum_{n=0}^{\infty} x^n \frac{(n+k)!}{k!n!}, \tag{A7}$$

which holds for $|x| < 1$, to the right-hand side of equation (A 6) with $n = M'$, $k = m$ and $x = a(1 - 1/N)$, we obtain

$$\hat{p}_m = \left(\frac{1}{(a^{-1} - 1)N}\right)^m \left(1 + \frac{1}{(a^{-1} - 1)N}\right)^{-m-1}. \tag{A8}$$

Substituting this equation into equation (A 3) and summing with respect to $m$, we finally obtain the Hassell model (2.8).

## A.2. Derivation of equation (3.3)

Model (3.3) can be simply derived from equation (3.1) using a probability-generating function. Note that equation (3.1) can be rewritten as

$$f(N) = E\left[\lambda N[1 + (e^{u/\bar{R}} - 1)N]^{-s}\right], \tag{A9}$$

where $E[A(s)]$ represents the expected value of a function $A(s)$ of a random variable $s$. This equation can be rewritten as

$$f(N) = \lambda N\, G\left([1 + (e^{u/\bar{R}} - 1)N]^{-1}\right), \tag{A10}$$

where $G(z) \equiv E[z^s]$ is the probability-generating function of $s$. As equation (3.2) shows, $s - s_0$ follows the

negative binomial distribution with mean $\delta$ and shape parameter $k$, and its generating function is known to be

$$E[z^{s-s_0}] = \left(1 + (1-z)\frac{\delta}{k}\right)^{-k}. \tag{A 11}$$

The generating function of $s$ is written as $G(z) = z^{s_0}E[z^{s-s_0}]$, and substituting equation (A 11) into the right-hand side of this equation yields

$$G(z) = z^{s_0}\left(1 + (1-z)\frac{\delta}{k}\right)^{-k}. \tag{A 12}$$

Applying this function to equation (A 10) finally gives model (3.3).

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
