## [Reviewer comments · Royal Society Open Science]

Review History

RSOS-182178.R0 (Original submission)

Review form: Reviewer 1

Is the manuscript scientifically sound in its present form?

Yes

Are the interpretations and conclusions justified by the results?

Yes

Is the language acceptable?

Yes

Is it clear how to access all supporting data?

Not Applicable

Do you have any ethical concerns with this paper?

No

Have you any concerns about statistical analyses in this paper?

No

Recommendation?

Major revision is needed (please make suggestions in comments)

Comments to the Author(s)

This paper derives the Hassell model of population dynamics from an underlying resource competition model, and thereby links the exponent of the Hassell equation to the inequality of individuals in competition (i.e., contest versus scramble). The basic model is a 'cartoon model' with restricted realism, but this is not a disadvantage; on the one hand, no one expects a complex realistic situation to be described by as a simple model as the Hassell equation, and on the other hand, the simplicity of the model makes it easy to discern the mechanism controlling competitive inequality, which then will operate also in much more complex situations. In the second half of the paper, the author also considers two extensions to the basic 'cartoon' model. Another nice feature is that the model links the Hassell and Ricker equations in a mechanistic way.

I do have, however, two conceptual difficulties with the model.

(1) Deterministic models of population dynamics, such as the Hassell model, apply in the limit of infinitely large populations. In contrast, the binomial distribution in equation (2.1) assumes a finite population. It is not clear how to resolve this discrepancy. What happens if one takes $N, M \rightarrow \infty$ in (2.1), i.e., a Poisson distribution instead of a binomial?

(2) Assuming a variable amount of resources is a key step in recovering the Hassell model [and not a "slight change" as said in line 95]. It is not clear what this variation in the total resource means; one population has only one total resource abundance at a time, and the author clearly does not mean temporal variation. The averaging in equation 2.7 is consistent with the scenario where the population is divided into groups of exactly N individuals each, and each group is given a random amount of resources. Such a subdivision could also solve the first problem described above. There is, however, no mention of any population subdivision in the ms. Note that subdivision has been invoked to explain overcompensation.

Minor comments:

- to improve readability, please start a new paragraph for each topic. For example, the last paragraph of the Discussion includes three largely unrelated topics, an alternative model, the interpretation of the exponent of the current model, and its extension to several species.
- exact compensation means that $N(t+1)$ is a constant independent of $N(t)$. What the Introduction calls exact compensation [such as the Beverton-Holt model] was called undercompensation in the literature.
- the distribution of resources in reality need not be exponential only because we assume to have no more information than the mean. Hence I find section 2c not convincing, but on the other hand I have no objection against assuming a particular [i.e., exponential] distribution.

Review form: Reviewer 2 (Juniper Simonis)**Is the manuscript scientifically sound in its present form?**

Yes

Are the interpretations and conclusions justified by the results?

Yes

Is the language acceptable?

Yes

Is it clear how to access all supporting data?

Not Applicable

Do you have any ethical concerns with this paper?

No

Have you any concerns about statistical analyses in this paper?

No

Recommendation?

Accept with minor revision (please list in comments)

Comments to the Author(s)

[See attached file (Appendix A)]

Decision letter (RSOS-182178.R0)

12-Apr-2019

Dear Dr Anazawa,

The editors assigned to your paper ("Inequality in resource allocation and population dynamics models") have now received comments from reviewers. We would like you to revise your paper in accordance with the referee and Associate Editor suggestions which can be found below (not including confidential reports to the Editor). Please note this decision does not guarantee eventual acceptance.

Please submit a copy of your revised paper before 05-May-2019. Please note that the revision deadline will expire at 00.00am on this date. If we do not hear from you within this time then it will be assumed that the paper has been withdrawn. In exceptional circumstances, extensions may be possible if agreed with the Editorial Office in advance. We do not allow multiple rounds of revision so we urge you to make every effort to fully address all of the comments at this stage. If deemed necessary by the Editors, your manuscript will be sent back to one or more of the original reviewers for assessment. If the original reviewers are not available, we may invite new reviewers.

When submitting your revised manuscript, you must respond to the comments made by the

referees and upload a file "Response to Referees" in "Section 6 - File Upload". Please use this to document how you have responded to the comments, and the adjustments you have made. In order to expedite the processing of the revised manuscript, please be as specific as possible in your response.

- Data accessibility

If you wish to submit your supporting data or code to Dryad (<http://datadryad.org/>), or modify your current submission to dryad, please use the following link:
<http://datadryad.org/submit?journalID=RSOS&manu=RSOS-182178>

- Competing interests

- Authors' contributions

- Acknowledgements

- Funding statement

on behalf of Dr Jose Carrillo (Associate Editor) and Kevin Padian (Subject Editor)
 openscience@royalsociety.org

Editor comments:

The reviewers are generally positive about your manuscript but offer a variety of suggestions. Could you please revise, taking into account all of these, and resubmit to us? Thanks for your submission

Comments to Author:

Reviewers' Comments to Author:

Reviewer: 1

Comments to the Author(s)

This paper derives the Hassell model of population dynamics from an underlying resource competition model, and thereby links the exponent of the Hassell equation to the inequality of individuals in competition (i.e., contest versus scramble). The basic model is a 'cartoon model' with restricted realism, but this is not a disadvantage; on the one hand, no one expects a complex realistic situation to be described by as a simple model as the Hassell equation, and on the other hand, the simplicity of the model makes it easy to discern the mechanism controlling competitive inequality, which then will operate also in much more complex situations. In the second half of the paper, the author also considers two extensions to the basic 'cartoon' model. Another nice feature is that the model links the Hassell and Ricker equations in a mechanistic way.

I do have, however, two conceptual difficulties with the model.

(1) Deterministic models of population dynamics, such as the Hassell model, apply in the limit of infinitely large populations. In contrast, the binomial distribution in equation (2.1) assumes a finite population. It is not clear how to resolve this discrepancy. What happens if one takes $N, M \rightarrow \infty$ in (2.1), i.e., a Poisson distribution instead of a binomial?

(2) Assuming a variable amount of resources is a key step in recovering the Hassell model [and not a "slight change" as said in line 95]. It is not clear what this variation in the total resource means; one population has only one total resource abundance at a time, and the author clearly does not mean temporal variation. The averaging in equation 2.7 is consistent with the scenario where the population is divided into groups of exactly N individuals each, and each group is given a random amount of resources. Such a subdivision could also solve the first problem

described above. There is, however, no mention of any population subdivision in the ms. Note that subdivision has been invoked to explain overcompensation.

Minor comments:

- to improve readability, please start a new paragraph for each topic. For example, the last paragraph of the Discussion includes three largely unrelated topics, an alternative model, the interpretation of the exponent of the current model, and its extension to several species.
- exact compensation means that $N(t+1)$ is a constant independent of $N(t)$. What the Introduction calls exact compensation [such as the Beverton-Holt model] was called undercompensation in the literature.
- the distribution of resources in reality need not be exponential only because we assume to have no more information than the mean. Hence I find section 2c not convincing, but on the other hand I have no objection against assuming a particular [i.e., exponential] distribution.

Reviewer: 2

Comments to the Author(s)
[See attached file]

Author's Response to Decision Letter for (RSOS-182178.R0)

See Appendix B.

RSOS-182178.R1 (Revision)

Review form: Reviewer 1

Is the manuscript scientifically sound in its present form?

No

Are the interpretations and conclusions justified by the results?

No

Is the language acceptable?

Yes

Is it clear how to access all supporting data?

Not Applicable

Do you have any ethical concerns with this paper?

No

Have you any concerns about statistical analyses in this paper?

No

Recommendation?

Reject

Comments to the Author(s)

In my first review, I raised two concerns, (i) how a deterministic model is applied to a finite population, and (ii) what is the interpretation of the resource distribution in this model. The author's answer to both is that $N(t+1)$ is not meant to be population size in generation $t+1$ but the expectation of the (random) population size at $t+1$.

While this answer is technically correct, unfortunately it is not satisfactory. Population models like the Hassell model are used to predict population dynamics over a number of generations. For this natural purpose, the expected population size is insufficient; one cannot substitute the mean of $N(t+1)$ when predicting $N(t+2)$ etc. Importantly, to predict even the expectation of $N(t+2)$, one needs the full distribution of $N(t+1)$. Hence while formally the result is correct, it cannot be used as a model describing the dynamics of a finite population with variable resource abundance.

Review form: Reviewer 2 (Juniper Simonis)

Is the manuscript scientifically sound in its present form?

Yes

Are the interpretations and conclusions justified by the results?

Yes

Is the language acceptable?

Yes

Is it clear how to access all supporting data?

Not Applicable

Do you have any ethical concerns with this paper?

No

Have you any concerns about statistical analyses in this paper?

No

Recommendation?

Accept as is

Comments to the Author(s)

The author has done a good job addressing my concerns and that of the other reviewer I believe the manuscript is suitable for publication as is.

My one very minor comment is that the paragraph starting on 296 (which was split out from the multi-topic paragraph) is a bit short. I'm wondering if there is anything else worth noting here about the exponents, such as future research directions of interest, etc.

-Dr. Juniper L. Simonis

Decision letter (RSOS-182178.R1)

30-May-2019

Dear Dr Anazawa:

Manuscript ID RSOS-182178.R1 entitled "Inequality in resource allocation and population dynamics models" which you submitted to Royal Society Open Science, has been reviewed. The comments of the reviewer(s) are included at the bottom of this letter.

Please submit a copy of your revised paper before 22-Jun-2019. Please note that the revision deadline will expire at 00.00am on this date. If we do not hear from you within this time then it will be assumed that the paper has been withdrawn. In exceptional circumstances, extensions may be possible if agreed with the Editorial Office in advance. We do not allow multiple rounds of revision so we urge you to make every effort to fully address all of the comments at this stage. If deemed necessary by the Editors, your manuscript will be sent back to one or more of the original reviewers for assessment. If the original reviewers are not available we may invite new reviewers.

- Ethics statement

- Data accessibility

- Competing interests

- Authors' contributions

- Acknowledgements

- Funding statement

on behalf of Dr Jose Carrillo (Associate Editor) and Kevin Padian (Subject Editor)
openscience@royalsociety.org

Subject Editor comments:

Thank you for this revision. As you'll see, the reviewers offer a split view; however, I have made a decision on the basis of the reviews received to avoid delaying you unnecessarily. While I am concerned that the more negative reviewer does not consider your explanation to be sufficiently compelling, I am recommending that you are granted a further round of revision to amend the paper and provide a more thorough response to their concerns. Do note, however, that it is unusual for the journal to offer more than one round of revision, and we will not be able to consider a further round of revision and review if neither I nor the referee is satisfied once you have resubmitted.

Reviewer comments to Author:

Reviewer: 2

Comments to the Author(s)

The author has done a good job addressing my concerns and that of the other reviewer I believe the manuscript is suitable for publication as is.

My one very minor comment is that the paragraph starting on 296 (which was split out from the multi-topic paragraph) is a bit short. I'm wondering if there is anything else worth noting here about the exponents, such as future research directions of interest, etc.

-Dr. Juniper L. Simonis

Reviewer: 1

Comments to the Author(s)

In my first review, I raised two concerns, (i) how a deterministic model is applied to a finite population, and (ii) what is the interpretation of the resource distribution in this model. The author's answer to both is that $N(t+1)$ is not meant to be population size in generation $t+1$ but the expectation of the (random) population size at $t+1$.

While this answer is technically correct, unfortunately it is not satisfactory. Population models like the Hassell model are used to predict population dynamics over a number of generations. For this natural purpose, the expected population size is insufficient; one cannot substitute the mean of $N(t+1)$ when predicting $N(t+2)$ etc. Importantly, to predict even the expectation of $N(t+2)$, one needs the full distribution of $N(t+1)$. Hence while formally the result is correct, it cannot be used as a model describing the dynamics of a finite population with variable resource abundance.

Author's Response to Decision Letter for (RSOS-182178.R1)

See Appendix C.

RSOS-182178.R2 (Revision)

Review form: Reviewer 1

Is the manuscript scientifically sound in its present form?

Yes

Are the interpretations and conclusions justified by the results?

Yes

Is the language acceptable?

Yes

Do you have any ethical concerns with this paper?

No

Recommendation?

Accept with minor revision (please list in comments)

Comments to the Author(s)

We are in agreement with the author that his model predicts the expected, rather than the actual, population size in the next generation, and that this expectation cannot be used iteratively to predict population size in the subsequent generations. In my opinion, this severely limits the usefulness of the model, and it is a fact that must be pointed out and discussed. The current text correctly says that N_{t+1} is the expectation, but it does not explain the consequence that the model is not suitable for iterative predictions. Since most readers automatically take a discrete-time population model to be iterative, a departure from this assumption has to be discussed.

Further, line 77 attributes the difficulty only to demographic stochasticity, but in fact it is also due to taking expectation over the resource distribution. In line 75, "We assume" should read "We show".

Decision letter (RSOS-182178.R2)

28-Jun-2019

Dear Dr Anazawa:

On behalf of the Editors, I am pleased to inform you that your Manuscript RSOS-182178.R2 entitled "Inequality in resource allocation and population dynamics models" has been accepted for publication in Royal Society Open Science subject to minor revision in accordance with the referee suggestions. Please find the referees' comments at the end of this email.

The reviewers and Subject Editor have recommended publication, but also suggest some minor revisions to your manuscript. Therefore, I invite you to respond to the comments and revise your manuscript.

- Ethics statement

- Data accessibility

It is a condition of publication that all supporting data are made available either as supplementary information or preferably in a suitable permanent repository. The data accessibility section should state where the article's supporting data can be accessed. This section should also include details, where possible of where to access other relevant research materials

such as statistical tools, protocols, software etc can be accessed. If the data has been deposited in an external repository this section should list the database, accession number and link to the DOI for all data from the article that has been made publicly available. Data sets that have been deposited in an external repository and have a DOI should also be appropriately cited in the manuscript and included in the reference list.

If you wish to submit your supporting data or code to Dryad (<http://datadryad.org/>), or modify your current submission to dryad, please use the following link:
<http://datadryad.org/submit?journalID=RSOS&manu=RSOS-182178.R2>

- **Competing interests**

- **Authors' contributions**

- **Acknowledgements**

- **Funding statement**

Because the schedule for publication is very tight, it is a condition of publication that you submit the revised version of your manuscript before 07-Jul-2019. Please note that the revision deadline will expire at 00.00am on this date. If you do not think you will be able to meet this date please let me know immediately.

Kind regards,

on behalf of Dr Jose Carrillo (Associate Editor) and Kevin Padian (Subject Editor)
openscience@royalsociety.org

Subject Editor Comments to Author (Professor Kevin Padian):

Thanks for your submission. In your final version please make the correction requested by the reviewer, because it seems to be a very important point. If you disagree, please let us know. Best wishes.

Reviewer comments to Author:

Reviewer: 1

We are in agreement with the author that his model predicts the expected, rather than the actual, population size in the next generation, and that this expectation cannot be used iteratively to predict population size in the subsequent generations. In my opinion, this severely limits the usefulness of the model, and it is a fact that must be pointed out and discussed. The current text correctly says that N_{t+1} is the expectation, but it does not explain the consequence that the model is not suitable for iterative predictions. Since most readers automatically take a discrete-time population model to be iterative, a departure from this assumption has to be discussed.

Further, line 77 attributes the difficulty only to demographic stochasticity, but in fact it is also due to taking expectation over the resource distribution. In line 75, "We assume" should read "We show".

Author's Response to Decision Letter for (RSOS-182178.R2)

See Appendix D.

Decision letter (RSOS-182178.R3)

04-Jul-2019

Dear Dr Anazawa,

I am pleased to inform you that your manuscript entitled "Inequality in resource allocation and population dynamics models" is now accepted for publication in Royal Society Open Science.

Kind regards,
Alice Power

Editorial Coordinator
Royal Society Open Science
openscience@royalsociety.org

on behalf of Dr Jose Carrillo (Associate Editor) and Kevin Padian (Subject Editor)
openscience@royalsociety.org

Appendix A

This is a review of the manuscript entitled "Inequality in resource allocation and population dynamics models" by Masahiro Anazawa for Royal Society Open Science

This manuscript tackles the gap in knowledge between usage and derivation of the Hassell (generalized logistic) population model, where the usage interpretation of unequal resources had (before this manuscript) not yet been supported by a first-principles derivation. This manuscript provides that derivation, thereby closing the knowledge gap.

The manuscript is well written and structured and conveys a clear and concise message, and I support its publication. I have a few minor editing suggestions that I believe will improve the manuscript's efficacy and readability:

- I'm not entirely following the statement on lines 77 and 78 "When there are M resource units in total, the competition for each unit is repeated M times." Can you please explain this further in the text for clarification?
- Consider pulling the Hassell model written on line 15 out as its own equation not in text (make it be not in-line)
- On line 50, change "population is subject to a mortality linearly increasing with the density" to "population is subject to mortality that linearly increases with density"
- The explanation of the justification for relaxing the assumption on Line 95 is a little rough and could be smoothed out a little bit. I know that you get to it in more detail in a later section, but it seems to need more information or smoother writing.
- It's already been clearly shown (for example in Geritz and Kisdi 2004, which you cite) that the Ricker and Beverton-Holt Models are limiting cases of the Hassell Model. To that end, I don't think it's needed for the author to state this in the abstract, nor to devote a subsection (2b) within the main body to this.
- The paragraph of Section 2c (starts on line 131) needs a reordering of the sentences in its second half:

Delete the sentence starting "In general, the following..."

Then, starting at the sentence beginning "Consider a continuous variable..."

Follow with the sentence starting "As suggested by this example, the distribution that..." [Although the leading clause "As suggested by this example" should be removed from the sentence]

Then follow with the sentence starting "The distribution that maximizes..." and then the sentence starting "For example, when we know..."

And then include the sentence starting "When only..."

And then finish with the sentence starting "In this interpretation"

- On line 148, I would suggest replacing "However, in real populations..." with "However, in many real populations..."

- On line 188, I would suggest replacing "...can be a real number..." with "...can be any real number..."

- I would suggest using the "e" symbol with the exponent as an exponent, rather than "exp" and a parenthetical in Equation 3.8

- Start a new paragraph with the sentence beginning "Although this study..." on line 281.
- Consider adding a more general concluding paragraph to the end of the discussion, as presently the manuscript ends on discussion of things not addressed in the present manuscript (competition).

-Dr. Juniper L. Simonis

Appendix B

RESPONSE TO COMMENTS BY REVIEWER 1

MS Reference Number: RSOS-182178

MS Title: Inequality in resource allocation and population dynamics models

MS Author: Masahiro Anazawa

I am grateful to reviewer 1 for the critical comments and useful suggestions that have helped me to improve my paper considerably. As indicated in the responses that follow, I have taken all these comments and suggestions into account in the revised manuscript.

Comments by reviewer 1

Comment 1-1

(1) Deterministic models of population dynamics, such as the Hassell model, apply in the limit of infinitely large populations. In contrast, the binomial distribution in equation (2.1) assumes a finite population. It is not clear how to resolve this discrepancy. What happens if one takes $N, M \rightarrow \infty$ in (2.1), i.e., a Poisson distribution instead of a binomial?

Response

I agree with you that deterministic models of population dynamics apply in the limit of infinitely large populations. For finite populations, the population size at the next generation should deviate from the model's prediction due to demographic stochasticity. In this paper, I assume finite populations and that the Hassell model describes the expected population size at the next population as a function of the present population size. In this interpretation, the discrepancy you are concerned about does not occur. However, in the original manuscript, I had not stated this interpretation clearly. In the revised manuscript, I have added some sentences to explain this interpretation (lines 75–78). In the limit of $N \rightarrow \infty$ and $\bar{R} \rightarrow \infty$, the Hassell model (2.8) approaches a deterministic model.

Comment 1-2

(2) Assuming a variable amount of resources is a key step in recovering the Hassell model [and not a "slight change" as said in line 95]. It is not clear what this variation in the total resource means; one population has only one total resource abundance at a time, and the author clearly does not mean temporal variation. The averaging in equation 2.7 is consistent with the scenario where the population is divided into groups of exactly N individuals each, and each group is given a random amount of resources. Such a subdivision could also solve the first problem described above. There is, however, no mention of any population subdivision in the ms. Note that subdivision has been invoked to explain overcompensation.

Response

In the original manuscript, I failed to explain well the meaning of the distribution of total resource R . This distribution means a probability, rather than statistical, distribution. At each generation, R takes one value, but I assume that we don't know this value and only know its probability distribution density. In other words, at each generation, the value of R is randomly determined according to the probability density. What I calculate in equations (2.7) and (2.8) is the expected population size at the next generation that we can infer based on the knowledge we have (the probability density). In the revised manuscript, I have added some sentences to explain the distribution of R in section 2(a) (lines 101–107).

Several interpretations of this probability distribution is possible. I had described one interpretation

in section 2(c) from the viewpoint of the maximum information entropy principle in the original manuscript. Another possible interpretation is that, as you mentioned, R varies randomly over time following the exponential distribution, which was not mentioned before in the manuscript. I think this interpretation may be more acceptable to many readers, so I have added a paragraph that explains this interpretation at end of section 2(c). A subdivision of a population is also a possible interpretation, but I don't take this interpretation as I feel this is a little ecologically unnatural.

Minor comments:

Comment 1-3

- to improve readability, please start a new paragraph for each topic. For example, the last paragraph of the Discussion includes three largely unrelated topics, an alternative model, the interpretation of the exponent of the current model, and its extension to several species.

Response

I have divided the second paragraph in the Introduction section of the original manuscript into two paragraphs (line 33). I have also divided the third and the fourth paragraphs in the Discussion section into two (line 284) and three paragraphs (lines 296 and 300), respectively.

Comment 1-4

- exact compensation means that $N(t+1)$ is a constant independent of $N(t)$. What the Introduction calls exact compensation [such as the Beverton-Holt model] was called undercompensation in the literature.

Response

It is true that the Hassell model with $b=1$ (the Beverton-Holt model) shows undercompensating density dependence at small population sizes, but at large populations, $N(t+1)$ is almost constant independent of $N(t)$. Hence, it has exact compensating density dependence only at large population sizes. I have corrected a sentence explaining the curve for $b=1$ to match this fact (line 18--20).

Comment 1-5

- the distribution of resources in reality need not be exponential only because we assume to have no more information than the mean. Hence I find section 2c not convincing, but on the other hand I have no objection against assuming a particular [i.e., exponential] distribution.

Response

I agree with you that the distribution of resources in reality need not be exponential only because we assume to have no more information than the mean. However, the distribution I'm talking about here is a probability distribution rather than a statistical distribution. When we don't know anything about the statistical distribution but the mean value, the maximum information entropy principle shows that the most natural estimate of the distribution is an exponential one. I think your concern resulted from my poor explanation of the distribution of R . As I wrote in Response to Comment 1-2, I have added some sentences to explain this distribution in section 2(a) (lines 101–107). Further, I have added a paragraph to explain another interpretation of the distribution in section 2(c).

RESPONSE TO COMMENTS BY REVIEWER 2

MS Reference Number: RSOS-182178

MS Title: Inequality in resource allocation and population dynamics models

MS Author: Masahiro Anazawa

I am grateful to reviewer 2 for the useful suggestions that have helped me to improve my paper considerably. As indicated in the responses that follow, I have taken all these suggestions into account in the revised manuscript.

Comments by reviewer 2

Comment 2-1

- I'm not entirely following the statement on lines 77 and 78 "When there are M resource units in total, the competition for each unit is repeated M times." Can you please explain this further in the text for clarification?

Response

I am sorry that my explanation was not good. I think the connection before and after this sentence was bad, so I have deleted it and changed the next sentence to "Assuming that there are M resource units in total (i.e. the total resource amount is $R=uM$) and that each resource unit is consumed by a randomly selected individual," (lines 83–84).

Comment 2-2

- Consider pulling the Hassell model written on line 15 out as its own equation not in text (make it be not in-line)

Response

I have put the equation on a new line as suggested (line 16).

Comment 2-3

- On line 50, change "population is subject to a mortality linearly increasing with the density" to "population is subject to mortality that linearly increases with density"

Response

Thank you for your suggestion. I have corrected this part as suggested (line 53).

Comment 2-4

- The explanation of the justification for relaxing the assumption on Line 95 is a little rough and could be smoothed out a little bit. I know that you get to it in more detail in a later section, but it seems to need more information or smoother writing.

Response

I have replaced two sentences on lines 95--97 in the original manuscript with new sentences that explain the situation underlying the exponential distribution of resource introduced there (lines 101--107).

Comment 2-5

- It's already been clearly shown (for example in Geritz and Kisdi 2004, which you cite) that the Ricker and Beverton-Holt Models are limiting cases of the Hassell Model. To that end, I don't

think it's needed for the author to state this in the abstract, nor to devote a subsection (2b) within the main body to this.

Response

It is true that the Ricker and Beverton-Holt models had already been shown to be limiting cases of a Hassell model, but they have never been discussed clearly from the viewpoint of inequality in resource allocation. In this paper, these models are discussed from this viewpoint. So I would like to leave subsection 2(b). I have modified the sentence about these models in the abstract so that it mention their relationship to resource inequality.

Comment 2-6

- The paragraph of Section 2c (starts on line 131) needs a reordering of the sentences in its second half:

Delete the sentence starting "In general, the following..."

Then, starting at the sentence beginning "Consider a continuous variable..."

Follow with the sentence starting "As suggested by this example, the distribution that..."

[Although the leading clause "As suggested by this example" should be removed from the sentence]

Then follow with the sentence starting "The distribution that maximizes..." and then the sentence starting "For example, when we know..."

And then include the sentence starting "When only..."

And then finish with the sentence starting "In this interpretation"

Response

Thank you for your concrete suggestion. I have reordered these sentences as suggested, where some words have been added or deleted to ensure consistency (lines 142–153). The paragraph is much better now.

Comment 2-7

- On line 148, I would suggest replacing "However, in real populations..." with "However, in many real populations..."

Response

I have modified this part as suggested (line 164).

Comment 2-8

- On line 188, I would suggest replacing "...can be a real number..." with "...can be any real number..."

Response

I have modified this part as "... can be any positive real number ..." considering that the parameter must be positive (line 204). To ensure consistency, I have also replaced "an integer" with "a positive integer". (lines 203--204).

Comment 2-9

- I would suggest using the "e" symbol with the exponent as an exponent, rather than "exp" and a parenthetical in Equation 3.8

Response

I have modified the equation as suggested (line 207).

Comment 2-10

- Start a new paragraph with the sentence beginning "Although this study..." on line 281.

Response

I have started a new paragraph as suggested (line 300). Following a comment by reviewer 1 (comment 1-3), I have divided the fourth paragraph in the Discussion section of the original manuscript into three paragraphs including the one you suggested.

Comment 2-11

- Consider adding a more general concluding paragraph to the end of the discussion, as presently the manuscript ends on discussion of things not addressed in the present manuscript (competition).

Response

I have added a concluding paragraph to the end of the Discussion section as suggested (lines 307--311).

Appendix C

RESPONSE TO COMMENTS BY REVIEWERS

MS Reference Number: RSOS-182178.R1

MS Title: Inequality in resource allocation and population dynamics models

MS Author: Masahiro Anazawa

I am grateful to the two reviewers for the helpful comments and for taking the time and effort necessary to review the manuscript.

Comments by reviewer 2

The author has done a good job addressing my concerns and that of the other reviewer. I believe the manuscript is suitable for publication as is.

My one very minor comment is that the paragraph starting on 296 (which was split out from the multi-topic paragraph) is a bit short. I'm wondering if there is anything else worth noting here about the exponents, such as future research directions of interest, etc.

Response

Thank you for your positive comment and suggestion. I have added some text about some ideas that might help to derive Hassell models with real number exponents in the future (lines 299–301).

Comments by reviewer 1

In my first review, I raised two concerns, (i) how a deterministic model is applied to a finite population, and (ii) what is the interpretation of the resource distribution in this model. The author's answer to both is that $N(t+1)$ is not meant to be population size in generation $t+1$ but the expectation of the (random) population size at $t+1$.

While this answer is technically correct, unfortunately it is not satisfactory. Population models like the Hassell model are used to predict population dynamics over a number of generations. For this natural purpose, the expected population size is insufficient; one cannot substitute the mean of $N(t+1)$ when predicting $N(t+2)$ etc. Importantly, to predict even the expectation of $N(t+2)$, one needs the full distribution of $N(t+1)$. Hence while formally the result is correct, it cannot be used as a model describing the dynamics of a finite population with variable resource abundance.

Response

I understand that for finite populations, discrete-time population models like the Hassell model cannot be used to predict population dynamics over a number of generations, but the Hassell models derived in this paper are not intended at all to be applied repeatedly over multiple generations. Predicting population dynamics over multiple generations is not the sole use of discrete-time population models. They are also used as a tool to statistically understand the relationship between the population sizes at one generation and the next for the purpose of discussing the effect of intraspecific competition. For example, in the first half of Hassell (1975), he proposed his model as a mathematical expression to describe the density effect based on experimental data of finite populations. This paper considers Hassell models from this point of view. Hence, the output of the function, which expresses the expectation of the population size at the next generation, is not used as an input to the function. Therefore, the concerns you raised do not occur in this paper.

Appendix D

RESPONSE TO COMMENTS

MS Reference Number: RSOS-182178.R2

MS Title: Inequality in resource allocation and population dynamics models

MS Author: Masahiro Anazawa

I am grateful to the reviewer and the editors for the helpful comments on the previous version of my manuscript.

Subject Editor Comments to Author (Professor Kevin Padian):

Comment 1

Thanks for your submission. In your final version please make the correction requested by the reviewer, because it seems to be a very important point. If you disagree, please let us know. Best wishes.

Response

Thank you for your comments. I have made all the corrections requested by the reviewer. I have also added the following ethics statement just to be sure (line 315):

"Ethics. This work does not require an ethical assessment."

Reviewer comments to Author (Reviewer 1):

Comment 2

We are in agreement with the author that his model predicts the expected, rather than the actual, population size in the next generation, and that this expectation cannot be used iteratively to predict population size in the subsequent generations. In my opinion, this severely limits the usefulness of the model, and it is a fact that must be pointed out and discussed. The current text correctly says that N_{t+1} is the expectation, but it does not explain the consequence that the model is not suitable for iterative predictions. Since most readers automatically take a discrete-time population model to be iterative, a departure from this assumption has to be discussed.

Response

Thank you for the helpful comments. I also think this point is very important. I have added the following text to explain the limitation of the derived model at the end of section 2(a) (lines 123–125):

"Note that the model (2.8) predicts the expected, rather than the actual, population size in the next generation, so that it cannot be used iteratively to predict population sizes in the subsequent generations."

Comment 3

Further, line 77 attributes the difficulty only to demographic stochasticity, but in fact it is also due to taking expectation over the resource distribution.

Response

This sentence was inaccurate. I have removed the sentence (line 77).

Comment 4

In line 75, "We assume" should read "We show".

Response

I have changed "We assume" to "We show" as suggested (line 75).